# Computed Tomography Attenuation of Three-Dimensional (3D) Printing Materials—Depository to Aid in Constructing 3D-Printed Phantoms

**DOI:** 10.3390/mi14101928

**Published:** 2023-10-14

**Authors:** Yuktesh Kalidindi, Aravinda Krishna Ganapathy, Yash Nayak, Anusha Elumalai, David Z. Chen, Grace Bishop, Adrian Sanchez, Brian Albers, Anup S. Shetty, David H. Ballard

**Affiliations:** 1School of Medicine, Saint Louis University, St. Louis, MO 63104, USA; yuktesh.kalidindi@health.slu.edu; 2School of Medicine, Washington University in St. Louis, St. Louis, MO 63110, USA; aganapathy@wustl.edu (A.K.G.); nayakyr@wustl.edu (Y.N.); david.chen@wustl.edu (D.Z.C.); 3Mallinckrodt Institute of Radiology, Washington University School of Medicine, St. Louis, MO 63110, USA; anushae@wustl.edu (A.E.); bishop.g@wustl.edu (G.B.); adrians@wustl.edu (A.S.); anup.shetty@wustl.edu (A.S.S.); 4St. Louis Children’s Hospital Medical 3D Printing Center, BJC Healthcare, St. Louis, MO 63110, USA; brian.albers@bjc.org

**Keywords:** 3D printing, FormLabs, attenuation, stereolithography, computed tomography

## Abstract

Three-dimensionally printed phantoms are increasingly used in medical imaging and research due to their cost-effectiveness and customizability, offering valuable alternatives to commercial phantoms. The purpose of this study was to assess the computed tomography (CT) attenuation characteristics of 27 resin materials from Formlabs, a 3D printing equipment and materials manufacturer. Cube phantoms (both solid and hollow constructions) produced with each resin were subjected to CT scanning under varying tube current–time products with attenuation measurements recorded in Hounsfield units (HU). The resins exhibited a wide range of attenuation values (−3.33 to 2666.27 HU), closely mimicking a range of human tissues, from fluids to dense bone structures. The resins also demonstrated consistent attenuation regardless of changes in the tube current. The CT attenuation analysis of FormLabs resins produced an archive of radiological imaging characteristics of photopolymers that can be utilized to construct more accurate tissue mimicking medical phantoms and improve the evaluation of imaging device performance.

## 1. Introduction

In recent years, 3D printing has become increasingly used in medicine due to its versatility, offering a wide range of applications and customizability. The additive manufacturing workflow offers efficient prototyping capabilities, saving substantial time compared to traditional manufacturing methods [1]. The use of 3D printing in medical imaging has added to this technology’s utility as the volume rendering of patient-specific anatomy can be translated into a physical model. Three-dimensional printing capabilities also continue to become more versatile and accurate. Anatomical models of bone and vasculature obtained from CT scans can be 3D-printed with sub-millimeter precision [2,3]. Some 3D printers can produce models with varying materials with different mechanical properties [4]. These recent advances have enabled 3D printing technology to accurately represent radiological imaging.

Three-dimensionally printed phantoms are increasingly used in medical imaging and research due to their cost-effectiveness and customizability, offering valuable alternatives to commercially available phantoms. Since these phantoms often serve as a tool for imaging calibration, the performance of the 3D-printed material during radiological imaging such as computed tomography (CT) is critical. During a CT scan, X-rays pass through the object being analyzed, while a specific proportion is absorbed or scattered depending on both object density and material composition. This degree of X-ray impedance is referred to as attenuation and is quantitatively represented through Hounsfield units. Matching the attenuation of the 3D-printed phantom to the human tissues it is meant to represent allows for more accurate replication and performance of the model, which improves its overall functionality.

A wide array of 3D printing materials and technologies can be employed in the development of tissue-mimicking phantoms. The most widely used method for 3D printing is fused deposition modeling (FDM). During this process, thermoplastic filament is melted and extruded through a nozzle to build a 3D-printed product layer by layer [5]. It is a more affordable and accessible method of 3D printing, but has several deficiencies when compared to stereolithography (SLA) in the context of generating 3D-printed phantoms. SLA printing involves a liquid resin that is repeatedly solidified through photopolymerization upon exposure to UV light. Each layer’s curing process is repeated sequentially until the final three-dimensional model is built [6,7,8]. FDM-printed objects typically have visible layers, or what has been described as a “surface roughness” or “stair-stepping effect,” which creates a rougher surface, whereas SLA-printed objects are typically smoother and more detailed [5]. This impacts the resolution achievable by FDM printing and makes SLA more attractive for biomedical devices such as these phantoms that require smaller dimensions and a higher degree of precision.

In order to construct accurate imaging phantoms through SLA printing, a thorough understanding of the attenuation properties of these resins is necessary. Although many manufacturers list the physical properties of materials such as Shore hardness or Young’s modulus, expected linear attenuation values measured by Hounsfield units (HU) are often not listed. The particular HU value for a material is dependent upon the electron density of the material as well as the CT peak kilovoltage (kVp) setting chosen [9,10]. This information is crucial for physicians’ and researchers’ informed selection of physiologically similar biomaterials. An illustration of the process to make a 3D-printed phantom is shown briefly in Figure 1.

Analyses of the medical imaging properties of commercially available SLA resins are sparse. Most reports on the attenuation properties of additive manufacturing materials are centered on FDM or other technologies, with little to no analysis of photopolymers [11,12,13,14,15,16,17,18]. To date, few studies have focused on many SLA printer resins and their attenuation properties [9,19,20]. The purpose of this study is to analyze the attenuation attributes of resins from an SLA printer manufacturer to identify resins that can accurately mimic the radiological properties of different bodily tissues under varying tube currents.

## 2. Materials and Methods

Cube phantoms with an edge length of 5.08 cm were provided by FormLabs (Somerville, MA, USA) in two variations: solid and hollow. FormLabs provided the cube constructs, but the data for the study were obtained independently by the authors and are under their control for publication. The hollow cubes had a simple matrix inside, whereas the solid cubes were completely filled with the appropriate resin material. Both cubes had 100% infill and a layer height of 100 microns. FormLabs provided two of each cube phantom type for each resin material being tested in this study. Thus, each resin material was analyzed using two solid and two hollow cube phantoms. The inclusion of both hollow and solid cubes allowed for analysis of potential attenuation differences when material density was adjusted. Given the diversity of tissue density in the body, realistic phantoms need to exhibit attenuation properties closely matching those of hollow and solid structures within the body.

The cubes were designed using Autodesk 3D Studio Max version 2024.1, a computer-aided design (CAD) software. The prototypes were first examined to ensure that they had a clean structure with sharp edges and were easy to print. The cube structure was intentionally chosen due to the ease with which it can be reciprocated with different printers in future studies. After examination, the final cubes were printed using a FormLabs SLA printer with each of the resin materials available in the inventory.

This study included 27 different resins of varying compositions from FormLabs, as outlined in Table 1. A set of two hollow and two solid phantom cubes were placed on the CT gantry along with a beaker of water for the HU reference standard, as shown in Figure 2 and Figure 3. Axial images were acquired on a Siemens (Erlangen, Germany) Biograph CT scanner at a constant tube voltage of 120 kVp and varying tube currents of 50, 100, 200, and 300 mAs. For each different tube current–time product (mAs), two study investigators placed three regions of interest (ROIs) on each cube, as shown in Figure 4. ROIs were selected to be as large as possible while avoiding the edges of the cube phantoms. The selection of each ROI was left to the discretion of study investigators, with dimensions ranging from 5 mm to 30 mm. All CT analysis and segmentation were performed using the Sectra (Linköping, Sweden) picture archival and communication system. Attenuation measurements in HU acquired by both readers were averaged and displayed in the graphs (Figure 5 and Figure 6). Attenuation values were averaged across all tube currents and cube types, both hollow and solid.

Statistical analyses were conducted using Rstudio version 2023.06.1 cufunctions package [21], and graphs were created using GraphPad Prism version 10.0.0 for Windows, GraphPad Software, Boston, MA, USA, www.graphpad.com, accessed on 12 July 2023. Comparisons between attenuation values were done using a one-way ANOVA, while interrater analysis between the two CT readers at varying tube currents was analyzed via intraclass correlation coefficient (ICC) values. ICC values between 0.75 and 0.9 were taken to represent good reliability, while values greater than 0.9 indicated excellent reliability.

## 3. Results

Hounsfield unit measurements of the analyzed resin materials at 120 kvP and varying tube currents are observed in Figure 5 and Figure 6. Values are demonstrated in table format in Table 1. The averaged attenuation across all tube currents and cube types ranged from −3.33 to 2666.27 HU. Tube currents at 50, 100, 200, and 300 mAs revealed attenuation that ranged from −3.74 to 2686.46 HU, −4.93 to 2658.13 HU, −1.90 to 2662.17, and −2.76 to 2668.62, respectively. Most resin materials resulted in an average Hounsfield value that fell between −4.93 and 210.47 HU. The photopolymer resins that exhibited values outside of this range were the glass-filled resins (rigid 10k and rigid 4000) and the dental crown resins (Perm Crown, Perm CB AZ B1, Temp Crown $ Bridge CZ). At a tube current of 200 mAs, Rigid 10k and Rigid 4000 exhibited average attenuations of 811.00 and 342.29 HU, respectively.

Average attenuations for Perm Crown, Perm CB AZ B1, and Temp Crown $ Bridge CZ were 2651.88, 2662.17, and 2627.50 HU, respectively. No significant changes or trends in attenuation were observed as the tube current was varied (*p* = 1, F = 3.353 × 10^−5^).

To visualize the variation in attenuation measurements acquired by both readers, the data collected by the raters were plotted against each other for each cube type, both hollow and solid (Figure 7). The ICC values for tube currents at 50 mAs, 100 mAs, 200 mAs, and 300 mAs were 0.9998941, 0.9999364, 0.9998341, and 0.9999283, respectively, which were all within the ‘almost perfect’ range (ICC = 0.81–1.00) (Table 2).

## 4. Discussion

The main objective of this study was to investigate the CT attenuation characteristics of 3D printing materials from Formlabs to establish a correlation between SLA resins and physiologic tissues that exhibit similar attenuation properties. The attenuation ranges observed across the varying tube currents demonstrated a broad physiological range. The overall unaveraged attenuation from all measured values ranged from −6.41 to 2704.50 HU.

The five photopolymers exhibiting the highest average Hounsfield values have attenuation values similar to human osseous tissues. The three dental crown resins (Perm Crown, Perm CB AZ B1, Temp Crown $ Bridge CZ) with average Hounsfield values within 2600 to 2700 HU most closely resembled dense bony tissue (cortical bone) and metal [22]. Rigid 10K and Rigid 4000 resembled the upper and lower limits of trabecular bone attenuation, respectively [23,24,25]. Many resins fall within the range of 100 to 200 HU: Dental Model V2, Teeth AZ, Denture Base LP, biomedical amber, biomedical clear, biomedical white, biomedical, black, Clear V4, High temp, model V3, CW, Tough 1500 and 2000, Grey V4 and Pro. These CT attenuations corresponded similarly to the HU values of a select few Formlabs resins (Tough 1500 and 2000, Grey V4, clear V4, High temp) reported in a previous study [18]. These resin materials exhibited CT attenuations within the same range as the HU of lumbar vertebrae with osteopenia or osteoporosis [26,27]. The resin materials analyzed within this study have similar characteristics to a broad range of osseous tissues, including pathological bony structures.

Nylon 12 consistently exhibited the lowest attenuation observed throughout this study, at a minimum of −6.41 HU. Consequently, this resin can accurately model water and other simple fluids commonly found in the human body [9,22]. The attenuation of CSF was exemplified by the average Hounsfield value observed for Nylon 11 at 13.58 HU. Other physiologic correlations include Flexible 80A and muscle/soft tissue, Flexible V1 and intracranial hemorrhage, and elastic 50A and blood [9,22].

Table 3 displays resins and their corresponding tissue or pathology mimics.

There were no significant changes or trends in attenuation observed as the tube current varied. However, this further supports the high level of reliability exhibited between the two raters scanning the resins multiple times, as shown in Figure 7.

The broad range of observed attenuation characteristics can be attributed to the variation in material properties of the analyzed resins. The observed HU of a given material is directly proportional to the linear attenuation coefficient of that material. For X-ray energies used in CT, this attenuation represents a combination of Compton scattering events and photoelectric absorption [28]. The contribution of each of these processes to the attenuation of X-rays is dependent on both the physical density and elemental composition of each resin material. CT attenuation is directly proportional to both the density and the atomic number of an element making up a material. As expected, higher Hounsfield values were exhibited by the high-density resins composed of materials with higher atomic numbers. Perm Crown (2678 HU), a ceramic-based material, is reported to have a density of 1.4 to 1.5 g/cm^3^, whereas Elastic 50A resin (39 HU), a material composed of acrylic compounds, is reported to have a density of 1.02 g/cm^3^. Ceramic materials are made of elements with higher atomic numbers compared to acrylates, resulting in more profound photoelectric absorption and attenuation [29]. The curing process of photopolymers also changes the molecular structures of a liquid resin material. Without knowing the exact elemental composition and physical density of a resin after photopolymerization, an estimate of its attenuation properties cannot be accurately achieved. Thus, the CT imaging of cube phantoms was necessary to properly ascertain the attenuation characteristics of resin materials.

The Formlabs resins analyzed in this study share radiological imaging characteristics with a wide variety of physiologic tissues (apart from fat and lung, which are generally in the −30 to −200 HU range). This catalog of attenuations can streamline the process of constructing 3D-printed phantoms. Once a patient’s anatomy is scanned and segmented, the right resin can be chosen to ensure that the printed phantom has similar attenuation properties to the real tissue. This is essential for phantoms to produce consistent results during imaging and calibration.

Repositories of 3D printing material attenuation characteristics like this study can ensure that the correct tissue-mimicking phantom is constructed for the correct application. Task-specific imaging phantoms can be constructed to develop imaging protocols and optimize device calibration in a more realistic environment than standard quality control phantoms. The complexity of human tissues is often poorly represented by commercial phantoms. The homogenous nature of many commercial phantoms results in the loss of important contextual features such as anatomical noise [18,30]. Additively manufactured tissue-mimicking phantoms have the capacity for customization that enables the incorporation of background texture into phantoms of complex structures such as glandular breast tissue and lung tissue. Fostering this exceptionally realistic imaging environment enhances the versatility of 3D-printed phantoms in image quality assessments and other facets of medical imaging research, such as imaging device calibration or dosage planning. Three-dimensionally printed phantoms can also be uniquely designed to be more modular. Removing or adding specific parts of a tissue-mimicking phantom could be incredibly useful, depending on imaging and medical training needs. Recent modular 3D-printed phantom developments include an abdomen phantom with removable organs and a liver pathology phantom with movable lesions [31,32]. A well-designed and constructed 3D-printed phantom can be just as functional as a commercial phantom for a fraction of the cost [33]. Thus, the customizable and cost-effective nature of additively manufactured phantoms characterizes them as a viable alternative to commercial phantoms.

Our study is not without limitations. The use of 2-inch/5.08 cm cube phantoms may not adequately represent the complexities of human anatomical structures. The distinction between hollow and solid cubes could influence attenuation measurements, and although averages were employed to offset this, nuances specific to each cube type might be missed. Lastly, imaging performed on a single vendor (Siemens Biograph CT scanner, Erlangen, Germany) and a constant tube voltage of 120 kVp could narrow the study’s applicability, as variations in equipment and settings might influence results.

## 5. Conclusions

The analysis of FormLabs resin attenuation expands the versatility of SLA technology for medical applications. This attenuation reference data for Formlabs resins covers a broad expanse of physiologic tissues, resulting in a valuable resource to facilitate the additive manufacturing of tissue-mimicking phantoms. The enhanced accuracy, customizability, and cost-effectiveness of 3D-printed tissue-mimicking phantoms make them a viable alternative to commercial phantoms. As research and development progresses in this field, the importance of developing materials with optimized CT attenuation properties needs to be emphasized. Innovations in 3D printing material development will continue to advance the utility of the technology within medicine.

## Figures and Tables

**Figure 1 micromachines-14-01928-f001:**
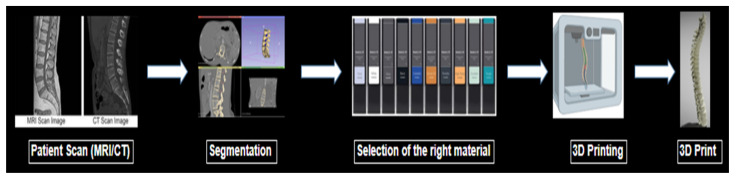
Illustration of the process to make a 3D-printed phantom.

**Figure 2 micromachines-14-01928-f002:**
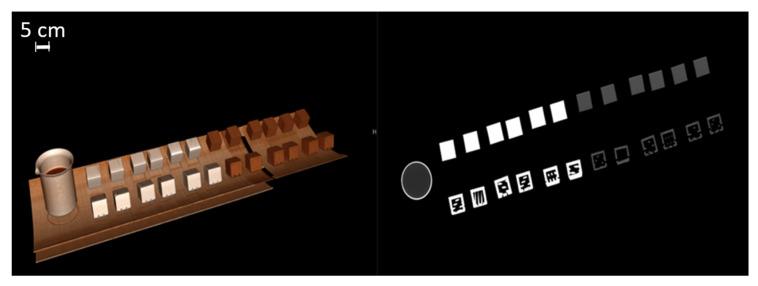
Three-dimensionally rendered reconstruction and oblique coronal reconstruction showing multiple cubes on CT gantry. A beaker of water serves as a Hounsfield unit reference standard (water set at 0 HU units).

**Figure 3 micromachines-14-01928-f003:**
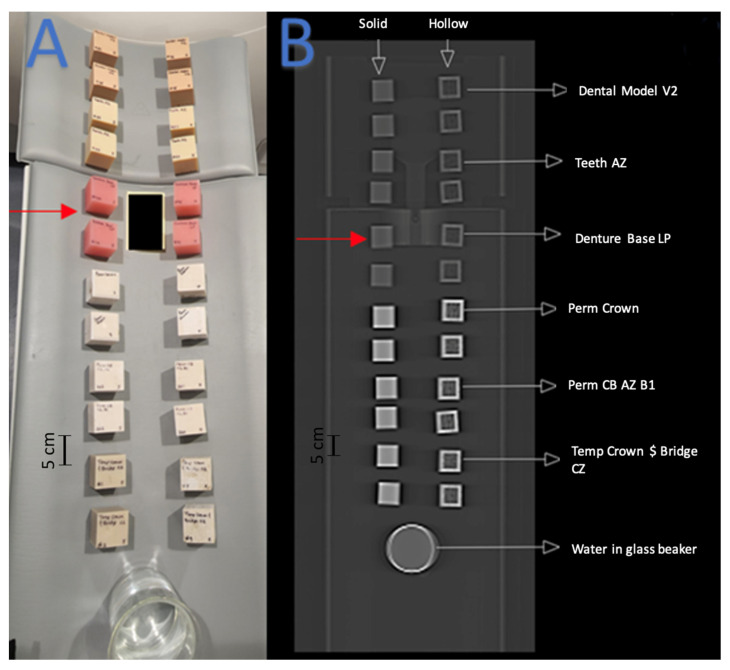
Red arrows indicate the set of cube phantoms made of Denture Base LP resin. (**A**) Arrangement of 3D-printed cube phantoms on CT gantry. (**B**) Coronal CT image of cube phantom gantry arrangement.

**Figure 4 micromachines-14-01928-f004:**
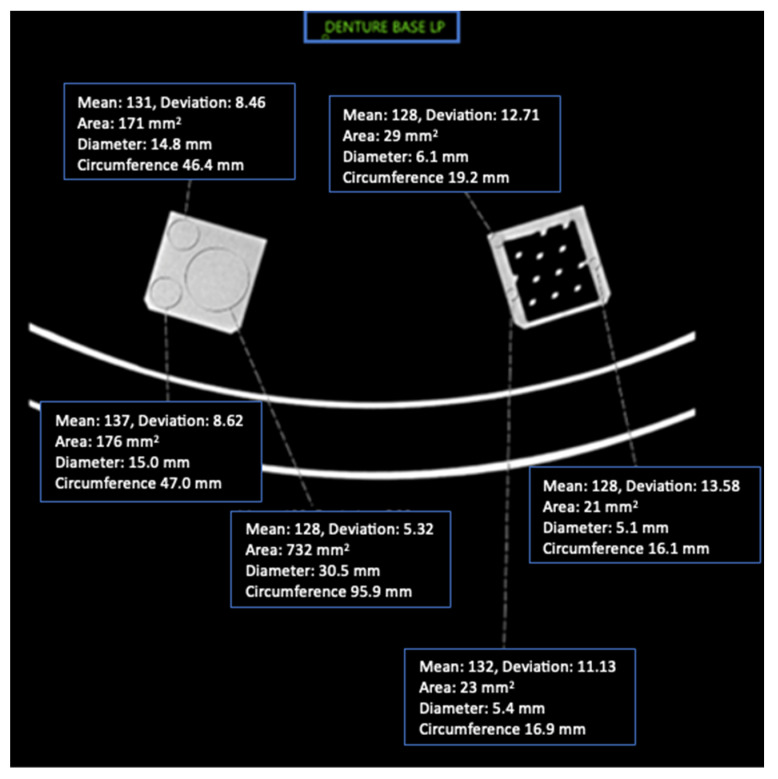
Axial CT image of Denture Base LP cube phantoms and ROI placement with imaging parameters.

**Figure 5 micromachines-14-01928-f005:**
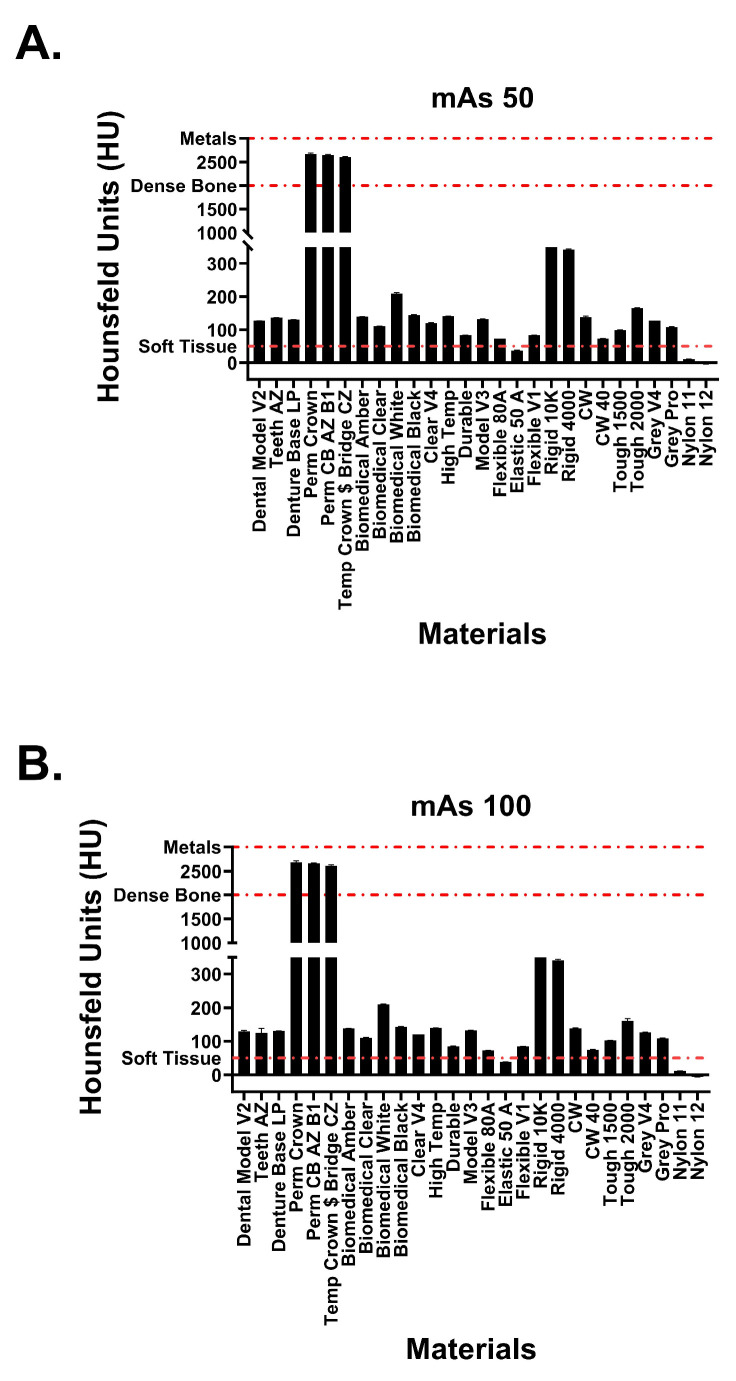
Attenuation values of Formlabs materials at tube currents of (**A**) 50 mAs and (**B**) 100 mAs. Red dotted lines signify attenuation values for physiological structures.

**Figure 6 micromachines-14-01928-f006:**
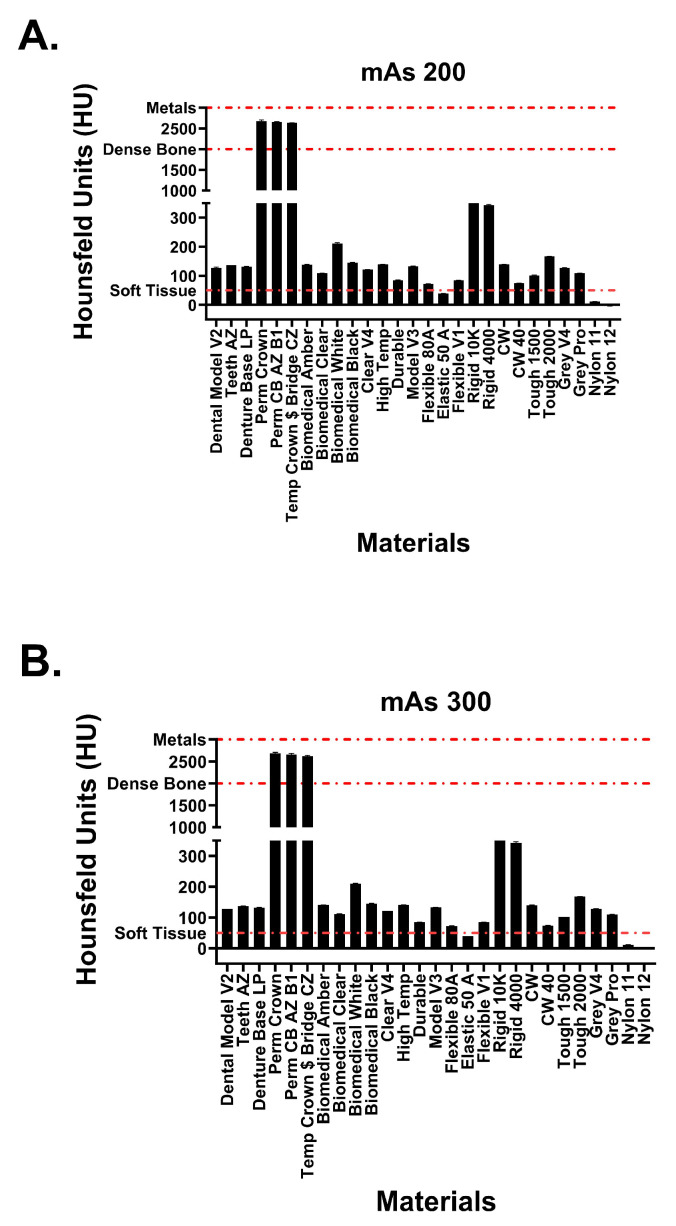
Attenuation values of Formlabs materials at tube currents of (**A**) 200 mAs and (**B**) 300 mAs. Red dotted lines signify attenuation values for physiological structures.

**Figure 7 micromachines-14-01928-f007:**
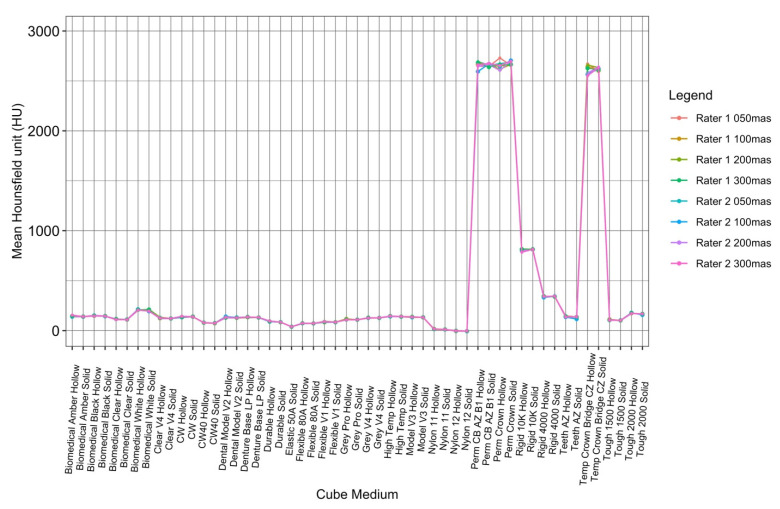
ICC interrater analysis of both readers at all tube currents.

**Table 1 micromachines-14-01928-t001:** Attenuation values (HU) of Formlabs materials at varying tube currents. Values represent the average between the scans of the solid and hollow constructs.

Materials	mAs = 50	mAs = 100	mAs = 200	mAs = 300	Average
Dental Model V2	127.17	129.415	127	128.085	127.9175
Teeth AZ	137.08	125.25	137.25	137.085	134.1663
Denture Base LP	131.085	130.585	132	132.17	131.46
Perm Crown	2671.33	2685.335	2675.25	2680.33	2678.061
Perm CB AZ B1	2651.25	2669.165	2655.667	2656.915	2658.249
Temp Crown $ Bridge CZ	2603.499	2612.249	2636.334	2621.669	2618.437
Biomedical Amber	139.665	138.915	139.335	140.8334	139.6871
Biomedical Clear	110.17	110.42	109.415	110.915	110.23
Biomedical White	209.915	209.765	211.085	209.585	210.0875
Biomedical Black	145.585	143.585	145.335	145.58	145.0213
Clear V4	120.335	120.67	121.335	121.67	121.0025
High Temp	140.5	139.75	139.5	140.58	140.0825
Durable	84.63	84.61	85.23	84.735	84.80125
Model V3	132.5	132.92	133	133.085	132.8763
Flexible 80A	72.14	72.035	71.72	72.45	72.08625
Elastic 50A	37.95	38.415	39.945	39.255	38.89125
Flexible V1	84.055	83.855	84.785	84.14	84.20875
Rigid 10K	813.17	812.08	813.585	813.835	813.1675
Rigid 4000	342.42	341.25	343.08	342.5	342.3125
CW	138.415	139.08	139.5	139.915	139.2275
CW 40	73.905	74.2	74.64	74.47	74.30375
Tough 1500	99.37	102.145	101.525	102.49	101.3825
Tough 2000	166.335	161.415	166.915	167.92	165.6463
Grey V4	127.585	127.08	127.585	128.25	127.625
Grey Pro	108.5	109	109	109.75	109.0625
Nylon 11	11.72	11.755	11.625	11.66	11.69
Nylon 12	−3.575	−6.075	−2.92	−4.16	−4.1825

**Table 2 micromachines-14-01928-t002:** ICC values quantifying interrater analysis of both readers at all tube currents.

Tube Current (mAs)	ICC Value
50 mAs	0.9998941
100 mAs	0.9999364
200 mAs	0.9998341
300 mAs	0.9999283

**Table 3 micromachines-14-01928-t003:** List of resins with close physiologic or pathologic correlates.

Resins	Mean HU	Tissue/Pathology Mimic	Reference
Perm Crown, Perm CB AZ B1, Temp Crown $ Bridge CZ	2651.582333	Dense bony tissue (cortical bone) and metal	[22]
Rigid 10k	813.1675	Upper limits of trabecular bone attenuation	[23,24,25]
Rigid 4000	342.3125	Lower limits of trabecular bone attenuation	[23,24,25]
Dental Model V2, Teeth AZ, Denture Base LP, biomedical amber, biomedical clear, biomedical white, biomedical black, Clear V4, High temp, model V3, CW, Tough 1500 and 2000, Grey V4 and Pro	135.6983	Lumbar vertebrae with osteopenia or osteoporosis	[26,27]
Nylon 12	−4.1825	Water and other simple fluids	[9,22]
Nylon 11	11.69	CSF	[22]
Flexible 80A	72.08625	Muscle/soft tissue	[9,22]
Flexible V1	84.20875	Intracranial hemorrhage	[9,22]
Elastic 50A	38.89125	Blood	[9,22]

## Data Availability

The data presented in this study are available in the article.

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
