# Peer review of "Computed Tomography Attenuation of Three-Dimensional (3D) Printing Materials—Depository to Aid in Constructing 3D-Printed Phantoms"

_micromachines, 2023, doi:10.3390/mi14101928_

Round 1

Reviewer 1 Report

I only have some minor comments:

1, what is ICC value? I think the authors may forget to define this.

2, In figure 2 and 3, there are no dimension information, or the scale bar is missing. In addition, it's difficult to read the texts in figure 3 b&c. 

3, I think the authors should also provide some info. regarding the printer and printing step of the cubes they assessed.

Reviewer 2 Report

Overall comments:

The subject matter of this manuscript deals the assessment of the CT attenuation characteristics of 27 different resin materials from FormLabs, which closely mimic a range of human tissues, enabling the construction of accurate, tissue-mimicking medical phantoms for enhanced medical training and surgical planning. The comprehensive analysis of FormLabs resin attenuation characteristics broadens the potential applications of Stereolithography (SLA) 3D printing in medicine, offering cost-effective and customizable alternatives to commercial phantoms. The paper was well-written and can be published after following minor concerns are revised.

Minor concerns:

1. In introduction, I recommend to describe the special advantages of producing tissue-mimicking phantoms using SLA printing compared to other printing technologies, including FDM. There is a lack of evidence as to why SLA, which uses ultraviolet rays, should be used rather than FDM, which is faster and can use a variety of materials.

2. The authors printed two two variations of cubes (solid and hollow). What was the reason for doing so? Describe the specific purpose of choosing that format to compare something.

3. Several typos, capitalization errors, abbreviation errors were found. Please check the whole manuscript again to revise the rest of them.

- Page 2 line 57 : Hounsfeld Units should be Hounsfield Units

- Caption on figure 4 ; Attenuation Values should be Attenuation values

- Caption on Table 1 : Attenuation Values (HU) should be Attenuation values

- Page 5 line 125 : The abbreviation of ICC must be described first.

none

Author Response

Please see the attachment. Thank you for your comments. 

Reviewer 3 Report

The paper presents a systematic work on the characterization of different materials for radiographic computed tomography (CT) signal attenuation. The paper could be of interest for in clinical testing.

Please modify the abstract indicating only the scientific issue to be investigated and the results. Avoid the use It as and introduction.

Figures 3 and 4 are too confusing, please split them whe possible.

Author Response

(The authors gave the same response as above.)

Reviewer 4 Report

The work “CT Attenuation of 3D Printing Materials—Depository to Aid in 2 Constructing 3D Printed Phantoms” by Yuktesh et al. assesses the computed tomography (CT) attenuation characteristics of all materials from a single 3D printing manufacturer by assessing Hounsfield Units (HU) measurements of standardized cube phantoms, since the 3D printed phantoms can be used in medical imaging and research due to their cost-effectiveness and customizability. They evaluated the attenuation characteristics of 27 different resin materials from FormLabs, using cube phantoms. The resins exhibited a wide range of attenuation values (-3.33 to 2666.27 HU), closely mimicking a range of human tissues from fluids to dense bone structures. The results are beneficial to enhanced medical training, evaluation of imaging device performance, and streamlined surgical planning. The paper is well written with abundant data, while the analysis is not comprehensive as the author claimed. The following issues should be addressed.

1.      In the introduction part, the authors should briefly explain the mechanism of CT attenuation by the interaction of material and X-ray.

2.      The quality of figures is quite low, and many texts and labels are not clear. Figures 2 and 3 should have scalebar.

3.      The readers cannot catch which cube is hollow and solid in the data, except Figure 5.

4.      Abbreviations like ICC should be provided before use.

5.      Most data were simply listed, and the physiologic or pathologic correlates were provided. However, it is better to explain a bit why specific resins have high or low HU values. Some physics-based explanation should be mentioned here.

Author Response

(The authors gave the same response as above.)

Round 2

Reviewer 3 Report

The Authors revised the paper according to suggestioni.